# Hand-Feel Touch Cues and Their Influences on Consumer Perception and Behavior with Respect to Food Products: A Review

**DOI:** 10.3390/foods8070259

**Published:** 2019-07-15

**Authors:** Ragita C. Pramudya, Han-Seok Seo

**Affiliations:** Department of Food Science, University of Arkansas, 2650 North Young Avenue, Fayetteville, AR 72704, USA

**Keywords:** hand-feel touch, haptics, tactile, cross-modal correspondence, sensory perception, consumer behavior, emotional response, packaging

## Abstract

There has been a great deal of research investigating intrinsic/extrinsic cues and their influences on consumer perception and purchasing decisions at points of sale, product usage, and consumption. Consumers create expectations toward a food product through sensory information extracted from its surface (intrinsic cues) or packaging (extrinsic cues) at retail stores. Packaging is one of the important extrinsic cues that can modulate consumer perception, liking, and decision making of a product. For example, handling a product packaging during consumption, even just touching the packaging while opening or holding it during consumption, may result in a consumer expectation of the package content. Although hand-feel touch cues are an integral part of the food consumption experience, as can be observed in such an instance, little has been known about their influences on consumer perception, acceptability, and purchase behavior of food products. This review therefore provided a better understanding about hand-feel touch cues and their influences in the context of food and beverage experience with a focus on (1) an overview of touch as a sensory modality, (2) factors influencing hand-feel perception, (3) influences of hand-feel touch cues on the perception of other sensory modalities, and (4) the effects of hand-feel touch cues on emotional responses and purchase behavior.

## 1. Introduction

Consumer perception and liking of a product are affected by both intrinsic (i.e., product-specific attributes such as sensory properties of a product) and extrinsic (i.e., external attributes that can be manipulated without intrinsically changing the product) cues [1,2,3,4]. For example, for fruits and vegetables typically presented without any packaging at retail stores, their sensory attributes such as appearance, aroma, and surface texture play an important role in consumer perception and liking, as well as purchase behavior during the point of sale. However, when fruits and vegetables are presented in opaque packages at retail stores, the consumer perception, liking, and decision making of the fruits and vegetables may be predominantly influenced by extrinsic packaging cues during the point of sale [5,6]. Because consumers are likely to categorize both a food item and its packaging taken together as a part of an overall product [7,8,9], information perceived and derived from food packaging may lead consumers to expect certain product sensory attributes and quality even before they consume it [10]. Packaging, therefore, is one of a number of important extrinsic cues that can affect consumer perception and liking of a product. In fact, most food and beverage products are now sold in a variety of packages at retail stores. 

Sense of touch plays an important role in consumer perception, evaluation, and decision making of a product during the point-of-sale transaction, product usage, and product consumption. Because of this role, consumers are more likely to prefer products when retailers allow them to appraise the products using their hands [11]. For many products, both touch and visual cues have been regarded as dominating consumers’ product experience throughout the entire cycle of product usage, i.e., from point of sale to usage cues [12]. In their book, Hultén et al. [13] emphasized the dominance of touch cues in sensory marketing: *“Seeing is reinforced by touch, in that touch helps us get a fuller understanding of what we see”* (p. 90). In other words, although during a point-of-sale transaction, most consumers typically rely on visual inputs to generate first impressions of a product, inputs from the sense of touch can provide confirmation of the initial visual impression, thereby creating a secondary impression of the product. Interestingly, touch cues exhibit a bidirectional effect with respect to the evaluation/appreciation of products. Touch cues reflect a positive effect in the evaluation of products that can be best explored by touching (e.g., a pillowcase or a washcloth) when the products are deemed of high quality, but they reflect a negative effect in the evaluation of low-quality products [14].

Since a sense of touch has historically provided a means of communication of positive or negative emotions [15,16], it is not surprising that touch cues derived or perceived from a food product or its packaging can elicit emotional responses when consumers explore or consume the product. In the presence of touch cues from a product, the perceived quality, performance, and usefulness of the product, as well as connotations associated with it, have been observed to evoke specific emotional responses to the product [17]. Consumer interaction with a product via touching could provide a sense of pleasure and comfort from a tangible object [18].

Touch cues derived from food products or their packaging, whether mouthfeel or hand-feel, may potentially help food industries enhance preference for, satisfaction with, and purchase intent with respect to products. Indeed, product packaging explored through touching has been increasingly recognized as an effective marketing tool [19], which is associated with rapidly-growing interest in the research related to product packaging design [20]. The close relationship between touch and emotions has also sparked research showing emotions evoked by food product or packaging. This increase and further growth of interest in such topics are kindled by recent discoveries that food-evoked emotions can predict consumer acceptance of products better than hedonic ratings of products [21,22,23].

While numerous studies and reviews have highlighted the fact that oral touch cues (e.g., mouthfeel) can modulate consumer perception and liking of products [24,25], surprisingly little is known about hand-feel touch cues and their influences on food perception, acceptance, and experience. This review will therefore provide (1) an overview of touch as a sensory modality, (2) factors affecting hand-feel perception, (3) effects of hand-feel touch cues on the perception of other sensory modalities, and (4) influences of hand-feel touch cues on emotional responses and purchase behavior in the context of food and beverage experience. Here, the food and beverage experience refers to consumer interaction with a food/beverage product from the point of sale to consumption. Background and knowledge gathered from this review will emphasize the importance of hand-feel touch cues on consumer perception and behavior during such an experience.

## 2. A Sense of Touch

### 2.1. Concept and Terminology

Although previous studies in a variety of fields have used “haptic” and “tactile” interchangeably to refer to perception through a sense of touch, they should not be characterized as meaning the same thing. More specifically, Sherrington [26] distinguished between haptic and tactile perceptions based on respective concepts of active and passive touches. In a similar vein, Gibson [27] also equated active touch with “haptic perception”, while passive or stationary touch was called “tactile perception” (also see Reference [28]).

Gunther and O’Modhrain [29] considered the term “haptic” to embody all aspects referring to the sense of touch. The “haptic system”, referring to the collective group of anatomical structures that contribute to the perception of haptic stimuli [29], allows us to perceive external stimuli through the sense of touch. Haptic sensations perceived through somatosensory receptors are categorized into two types: tactile sensation (or taction) and kinesthetic sensation (or proprioceptive sensation) (Figure 1). Tactile sensation, typically associated with the sensation of pressure, orientation, curvature, texture, thermal properties, puncture, and vibration [29], is perceived primarily through stimulation of the skin [30] where cutaneous receptors (mechanoreceptors and thermoreceptors) are located [31,32]. Kinesthetic sensation, associated with body position and movement, is perceived through stimulation to the kinesthetic receptors located in muscles, joints, and tendons [29,33,34]. Therefore, the term “tactile”, mediated only through cutaneous receptors, can be considered as a sub-category of “haptic”. For example, imagine that Olivia consumes popcorn using her right hand while holding the popcorn container in her left hand. When Olivia swirls, picks up, and then places pieces of popcorns into her mouth, she perceives haptic sensations of the popcorns from both cutaneous and kinesthetic receptors located in her right fingers, while she perceives tactile sensations of the container from cutaneous receptors placed in her left hand. Haptic perception, including kinesthetic perception (proprioception), can be more involved than tactile perception in the hand-feel touch perception of products. In fact, Gibson [27] demonstrated that participants achieved better perception of two-dimensional objects (e.g., cookie cutters) when they freely explored the shapes with their hands, thereby activating kinesthetic receptors, compared to when objects were statically placed in their hands and they only passively touched the objects.

### 2.2. Perception of Touch Cues

Sensory cues from touching (hereafter referred to as “touch cues”) can alert individuals of threats to their safety and well-being by the detection of temperature, vibrations, and weight information, while also informing them of the location of objects (spatial awareness) in their surroundings [35]. Processing of touch cues begins with stimulus detection on the skin that triggers the nervous system to deliver information to the spinal cord and relay it to the thalamus and the somatosensory cortex in the brain. The skin consists of multiple layers of tissues, with the epidermis comprising the first layer and the dermis directly located beneath. In the glabrous (hairless) skin (e.g., the fingertips), the intersecting boundary between the epidermis and dermis contains mechanoreceptors arranged to cause receptor activation [31]. The epidermis acts as a protective layer of tough dead cells for underlying layers, and it contains no blood supply [31]. Most sensory receptors are embedded in the dermis layer comprised of connective tissues and elastic fibers immersed in a semifluid and amorphous complex (referred to as a ground substance) [31,36]. A popular model of the physical properties of the skin characterizes the skin as a waterbed (“waterbed” model), imagined as “an elastic membrane enclosing an incompressible fluid” [31,37], and this model has been shown to satisfactorily fit with in vivo data [38].

Cutaneous receptors, located across the entire surface of the body (beneath both hairy and hairless parts), include mechanoreceptors (responsible for perceptions of pressure, slip, and vibration), thermoreceptors (for temperature perception), and nociceptors (for pain perception) [32]. There are four main types of mechanoreceptors: (1) slowly-adapting (SA) type I receptors (SA I; small receptive field) that end in Merkel cells, (2) slowly-adapting type II receptors (SA II; large receptive field) that end in Ruffling corpuscles, (3) fast-adapting (FA) type I afferents (FA I; small receptive field) that end in Meissner corpuscles, and (4) fast-adapting (FA) type II afferents (FA II; large receptive field) that end in Pacinian corpuscles [32,33]. The responses of these receptors to stimuli are dependent on two factors: (1) the receptive field size (i.e., the skin region in which the neurons can detect relevant signals) and (2) the relative adaptation rate (i.e., the rate at which the neurons adapt to a constant or static stimulus applied to the skin) [32]. Fast-adapting receptors first transmit impulses to the brain at the moment a stimulus is applied to the skin and then again when the stimulus is removed, while slowly-adapting receptors continue transmitting impulses as long as the stimulus is applied. Each of the four mechanoreceptors has its own features and functionalities. Merkel endings (SA I) play a role in (1) capturing information related to sustained pressure [39] and spatial deformation [40], (2) detecting very-low-frequency vibrations [41], (3) perceiving coarse textures [42], (4) detecting a pattern/form [43], and (5) manipulating a stable precision grasp [44]. Ruffini endings (SA II) serve to (1) detect high-frequency vibrations [41], (2) perceive fine textures [45], and (3) manipulate a stable precision grasp [44]. Meissner corpuscles (FA I) also manipulate a stable precision grasp by detecting low-frequency vibrations, thereby making them highly sensitive to dynamic impulses, but poorly sensitive to spatial recognition and static stimuli [41,44,46,47]. Finally, Pacinian corpuscles (FA II) receive information about sustained downward pressure, lateral skin stretching [48], and low dynamic sensitivity [39], and therefore play a role in (1) detecting the direction of object motion and force [49], (2) manipulating a stable precision grasp [44], (3) determining the finger position [50], and (4) detecting spatial deformation [40]. As described in the previous section, haptic sensations are classified into tactile and kinesthetic sensations. While the focus on this review is on tactile perception, it is worth noting that kinesthetic perception also plays a crucial role in daily life. Kinesthetic sensations refer to those that sense the position and movement of the body [34]. The primary receptors for these sensations are in the muscle spindle and Golgi tendon organs, which have been thought to contribute to the sense of limb position, movement, and position [34,51]. Besides the muscle spindles, joint receptors have been implicated in sensing joint movement, but are limited on signaling movement direction and joint position [51]. Additionally, the four mechanoreceptors contribute to the sense of movement, but the slowly-adapting cutaneous receptor Ruffini endings, in particular, can also sense limb position [34,52,53]. 

The other type of cutaneous receptors, thermoreceptors, can contribute to the perception of warmth and cold [54]. These sensations are mediated by a network of primary afferent nerve fibers, mainly C fibers and Aδ fibers, referred to as transient receptor potential (TRP) ion channels, that activate and react appropriately to environmental temperature [55,56]. In other words, these TRP channels, categorized into 7 families, are specialized to respond to specific ranges of temperatures and types of pain [56,57,58]. Of these 7 families, 3 are of particular interest in thermoreception: vanilloid TRP channels (TRPV), melastatin or long TRP channels (TRPM), and ankyrin transmembrane protein channels (TRPA) [56]. Warm sensations are generally transmitted by slowly-reacting unmyelinated C fibers, while cold sensations are mediated by faster-reacting myelinated Aδ fibers [59]; however, both types of fibers are responsible for the mediation of pain perception [56,59]. Before detailing the specific stimuli that could activate the specific thermoreceptive TRP channels, it must be noted that TRPVs 1 to 4 are activated in response to warm and high temperatures, while TRPA1 and TRPM8 are responsive to warm or cold sensations. TRPV1 responds to a wide variety of temperature and physical stimuli, i.e., temperatures in the approximate range 42–43 °C, capsaicin, inflammations, and neuropathic conditions [58,60]. Like TRPV1 that responds to high-temperature stimuli, TRPV2 also responds to relatively high temperatures, i.e., noxious heat (higher than 52 °C) [56]. TRPV3 and TRPV4 are activated by lower temperatures than TRPV1 and TRPV2, i.e., above 33 °C [61] and approximately between 24 and 34 °C [62,63], respectively. TRPA1 is activated by cold sensations, low temperatures, i.e., noxious cold (approximately 17 °C or below) [64], while TRPM8 is activated by temperatures below 26 °C [56,60]. 

In contrast with thermoreceptors that activate in specific temperature ranges, nociceptive afferents respond to both painful cold and hot stimuli [56]. In addition to responding to painful temperature stimuli, i.e., above approximately 40–45 °C and below 15 °C, nociceptors are also activated by other types of pain, such as intense pressure or actual (or potential) physical damage to the body [65]. With an anatomy similar to that of thermoreceptors, nociceptors are also composed of C fibers and Aδ fibers, and these two groups of receptors are extremely closely connected in terms of their activations [65]. In other words, a given stimulus, especially noxious heat or cold, could activate both thermoreceptors and nociceptors, but the range of stimuli for nociceptors extends to other actual (or potential) physical irritants to the body. Nociceptors are generally classified as mechano-nociceptors, polymodal nociceptors, and silent nociceptors according to their responsiveness to mechanical force, heat, and other exogenous irritants [66]. Mechano-nociceptors, primarily composed of Aδ fibers (type I Aδ fibers/thermal nociceptors and type II Aδ fibers/mechanoheat nociceptors), although C fibers are also involved, are responsive to stimuli creating moderate to excessive tissue damage by transmitting signals that increase in frequency with stimulus intensity [66,67]. Polymodal nociceptors, primarily composed of C fibers, respond to stimuli exerting intense mechanical deformation, diluted acid or other irritant chemical stimuli, and heating of the skin over 40 °C, and they have been reported to be sensitized to repeated stimuli [66]. Finally, silent nociceptors, composed of both Aδ and C fibers, are normally unresponsive to noxious stimuli except those of extreme intensity, and respond only when supporting tissues, i.e., skin, deep tissues, and joints, experience inflammation and post-stimulus injuries [66,68]. Upon contact with pain-creating stimuli, fast-conducting myelinated type I Aδ and II Aδ fibers are activated, initially resulting in painful sensations, while subsequent sustained painful sensations are caused by the activation of slow-conducting unmyelinated C fibers [67].

The skin can be categorized into three main types: glabrous (non-hairy sections of the human body), non-glabrous (hairy sections of the human body), and mucocutaneous (regions in the skin containing junctions at which mucous membranes transition to the skin) [69,70]. The glabrous skin contains all four types of mechanoreceptors (SA I, SA II, FA I, and FA II), while the hairy skin contains all except FA I (i.e., SA I, SA II, and FA II), instead containing fast-conducting myelinated Aβ fibers and slow-conducting unmyelinated C-tactile fibers [69,71,72,73]. Somatosensory receptors exhibit different degrees of sensitivity depending on skin type and location in the human body [31,74]. Different parts of the body and types of skin have shown varying degrees of touch sensitivity depending on the procedure used for measuring touch sensitivity. For example, Weinstein [75] reported that fingertips, followed by the upper lip, the cheeks, and the nose, to be the most sensitive areas when measured by a two-point discrimination task. In contrast, in a more recent study comparing touch sensitivities between the index fingertip and the tongue using the Semmes-Weinstein monofilaments, the tongue was found to be more sensitive than the index fingertip [76]. It should be noted that these studies have only considered the glabrous (i.e., non-hairy) and mucocutaneous parts of the body. While previous studies had generally agreed that glabrous sections are more sensitive than non-glabrous [77,78], when stimuli directly moved the hairs on the non-glabrous section of a human hand, the non-glabrous part was found to be more sensitive to air-puffs [79]. 

## 3. Factors Influencing Hand-Feel Touch Perception

Various factors influence the hand-feel touch perception of food and other materials [80]. Along with their independent influences on hand-feel touch perception, many of these factors interact with one another to contribute to the overall haptic perception or “feel” of an object [81]. There are, in general, three factors influencing hand-feel touch perception of food products: (1) product-related, (2) consumer-related (including physiological and psychological factors), and (3) external interface-related (e.g., container, tableware, cutlery, and packaging). 

### 3.1. Product-Related Factors

Much of the previous work investigating hand-feel touch perception has focused on fabric or paper samples. The term “fabric hand” is the common terminology used in the textile industry when describing the quality of fabric evaluated by hand touching [82]. When presented with a solid or semi-solid food product, humans naturally evaluate textural properties, such as firmness and deformation, using their sense of hand-feel touch. In a study evaluating the textural properties of puddings, bread, fruits, and vegetables using instrumental tools and human subjects, Szczesniak and Bourne [83] observed that untrained panelists actively touched the food products using their fingers and hands, whether directly touching the food products or indirectly by using cutlery items, when they were asked to judge the textural parameters of the food samples without eating them. In fact, the quality and ripeness of the fresh produce, such as fruits and vegetables, have traditionally been evaluated using hand-feel touch by consumers at retail stores, along with visual, auditory, and olfactory cues [84,85]. 

In recent years, there has been a surge of interest in eating with one’s hands, particularly in the restaurant industry [86]. The hand-feel touch perception of a food or beverage product is affected by its intrinsic product characteristics (i.e., its sensory attributes) that can be influenced by multiple factors that include ingredients, composition, physical structure, and processing methods. By hand touching, humans are likely to discern textural differences between samples that vary in composition, ingredients, and processing procedures [87,88,89,90,91,92]. For example, Pereira et al. [88] showed that cheese products varying in moisture content could be differentiated by hand touching; those with a lower moisture content were evaluated as firmer, curdier, and less sticky than those with higher moisture content. Another study showed that an ethnic flatbread (parotta) sample prepared with guar gum was rated higher with respect to hand-feel quality than a bread sample prepared with Arabic gum [90]. It should be also noted that hand-feel touch perception can be influenced by multisensory interactions with other sensory properties of a food or beverage product (for details, see Section 4.1, Section 4.2, Section 4.3, Section 4.4 and Section 4.5). 

### 3.2. Consumer-Related Factors

#### 3.2.1. Physiological and Demographic Factors

Skin temperature is of particular importance in hand-feel touch perception, with the skin and subdermal tissues extensively involved in the homeostatic regulation of body temperature [93]. Homeostatic regulation occurs by modifying blood flow through various skin tissues or through perspiration. Factors such as the tissue’s specific heat, its thermal conductivity, and the mass flow and temperature of blood induce variations in skin surface temperature, thereby affecting its vibratory sensitivity [94,95,96]. In addition to changes in vibratory sensitivity, varying skin temperatures can result in changes in fingertip roughness perception [97] and tactile spatial acuity [98]. Specifically, increasing skin temperature from 10 °C to 43 °C results in a notable increase in perceived roughness by the touch stimuli [97].

Individual demographics such as age and gender are considered to be another important factor influencing the hand’s touch perception, with aging found to influence touch/pressure sensitivity [99,100,101,102], vibrotactile sensitivity [103,104], and spatial acuity [105]. While some studies found older participants to be as good as younger participants with respect to tactile sensitivity [106], older participants have been found to exhibit a substantial decline in tactile sensitivity when measured using Semmes–Weinstein monofilaments [99,107]. Aging has also been found to decrease sensitivity to skin indentations (also a measure of tactile sensitivity) [100] and vibratory stimuli [103,104]. While the results from studies related to the effects of age on pain perception have widely varied depending on how the pain stimuli are induced, the consensus is that pain sensitivity, including thermal sensitivity, decreases with age [108,109,110]. Deterioration with respect to multiple touch sensations and capabilities has been theorized as being due to age-related changes in the skin’s mechanical properties, particularly the thinning of the dermis portion of skin and loss of dermal collagen, that result in increasingly inelastic and rigid skin tissues compared to those of younger individuals [100,111]. In addition, diminution of touch sensitivity has been considered as a result of a decrease in density, a change in the morphology of touch receptors, and/or an age-related increase in the frequencies of primary afferent neuropathies [100]. 

Women exhibit higher tactile sensitivity than men, probably due to their thinner skin resulting from hormonal conditioning [112]. Women have also been found to be more sensitive than men with respect to vibrotactile sensitivity [113], pressure sensitivity [75], thermal sensitivity [114,115], and pain sensitivity [115].

Physical dysfunction and health issues of individuals have also been found to influence touch perception. For example, female patients with rheumatic disease, in contrast to counterparts in a control group, exhibited lower tactile sensitivity [102]. A review of the effects of chronic pain on altered sensory perception concurred with the observation that, in general, individuals suffering from chronic pain experience a decreased in tactile-discrimination capability [116]. Frohlich and Meston [117] also reported that the finger-tactile sensitivity of women with sexual arousal disorder was associated with the disorder’s severity. Other physical impairments, such as blindness, could also influence perceptions of touch cues. With reduced sensitivity in one sense, impaired individuals have sometimes been shown to develop greater sensitivity and discriminatory ability in another specific sense [118]. For example, in a study comparing tactile sensitivities of blind, deaf, and unimpaired individuals, visually-impaired participants exhibited a greater tactile sensitivity than those in the other two groups [118]. Visually-impaired participants might naturally be expected to acquire greater sensitivity to touch cues in response to their loss of vision through habitual and repeated performance of important daily activities such as reading Braille texts [118]. Other studies have suggested that an increase in the tactile acuity of blind individuals is due not to their experience in performing certain activities requiring a sense of touch, but rather to visual impairment-induced “brain plasticity” [119,120]. While blind individuals may retain better tactile acuity throughout their lives, this capability, as for unimpaired individuals, declines with age [121,122].

#### 3.2.2. Psychological Factors

Specific emotional states [98,123] and chronic psychoemotional stress [124] have been found to impact hand-feel touch perception, and the effects of negative emotional states on tactile sensitivity vary depending on the type of emotion. More specifically, Kelly and Schmeichel [123] showed that the fear state decreases tactile sensitively, whereas the anger state has no effect, possibly explained by a three-dimensional model of emotion: valence, arousal, and motivation (approach versus avoidance). Although both fear and anger are categorized as negatively-valenced and high-arousal emotions, they differ in terms of motivational direction; while fear is associated with an avoidance motivation, anger is considered as an approach motivation. Thus, the difference in tactile sensitivity between anger and fear states may be interpreted in terms of motivational direction (approach versus avoidance). In addition, while an anger state has been found to increase finger temperature, a fear state has been observed to decrease finger temperature [125], leading to the modulation of tactile vibratory sensations [96]. The fear-induced finger-temperature decrease has been associated with reduced tactile sensitivity [98,123]. 

Individual motivation or preference to touch cues is another crucial factor influencing hand-feel touch perception. Individuals can be categorized as high or low autotelics using the “Need-for-Touch” (NFT) scale created by Peck and Childers [126] that measures personal motivation or preference to touch objects based on two sub-scales: instrumental and autotelic. Instrumental NFT measures a person’s tendency to touch related to a specific objective (e.g., to make a judgment for purchase; “*The only way to make sure a product is worth buying is to actually touch it*”). Autotelic NFT represents a person’s compulsivity or tendency to touch only for the sake of touching (e.g., “*Touching products can be fun*”). This scale has successfully been used to discern individual differences in perception based on different need-for-touch levels. For example, highly-autotelic individuals have been shown to more likely engage in a haptic exploration of a product because they feel a need to do so, and are more likely to be influenced by features that include a hedonic touch element [126,127]. Consumers often have a tendency to engage in impulsive behavior when a positively-affective reward is promised [128], and individuals exhibiting such tendencies are more inclined to touch a hedonic object [128]. A positive significant correlation between autotelicity and purchase intent has also been observed [126]. These findings suggest that highly-autotelic individuals would be more likely to engage in impulsive purchase behavior [129]. Krishna and Morrin [130] also showed that, depending on the individual NFT, non-diagnostic haptic cues such as a container’s textural impression, may not be as likely to influence perception and evaluations.

Autism spectrum disorder (ASD) has also been found to exhibit highly intense reactions (hyper-responsiveness) or reduced reactions (hypo-responsiveness) toward sensory cues such as touch [131]. Children with ASD have been shown to exhibit increased sensitivity to pressure pain and punctate sensation, suggesting abnormal feedback to touch stimuli [132]. Individuals with ASD also perceived lower pleasant-to-touch stimuli than those without ASD [133]. Individuals with alexithymia, another psychological condition (the inability to identify, describe, and interpret emotional states) [134], tend to experience heightened sensitivity to pressure-induced touch and pain [135,136]. These findings illustrate that certain health or mental conditions can affect an individual’s acceptance and perception with respect to a product assessment through a sense of touch. 

### 3.3. External Interface-Related Factors

#### 3.3.1. Container, Tableware, and Cutlery Items

Haptic qualities of food or beverage containers and cutlery items may affect a consumer’s haptic perception, especially texture perception, of the product contained within [137,138,139,140,141]. Schifferstein [141] examined experiences in drinking beverage samples from cups made from different materials with results showing the cup material significantly affected many attributes related to the drinking experience. For certain attributes, such as warmness, consumer ratings of a product attribute seemed to mimic ratings of container attributes. Tu et al. [142] also found that certain oral somatosensory sensations, e.g., cold perception, can be affected by the serving-cup material. This tendency for an individual to judge the product quality or acceptance in terms of one sensory modality in accordance with ratings based on another sensory modality has been referred to as “sensation transference” [143], “affective ventriloquism” [144], or “cross-modal correspondence” [145].

Numerous studies in the field of fabrics and apparel design have shown that different materials evoke different hand-feel sensations [146,147]. In addition, incorporation of fabrics and other reusable materials into reusable containers and tableware items has increased as consumers have become more concerned with reducing environmental impacts related to product purchase [148]. Since this may make consumers more willing to pay more for such products, it is unsurprising that companies are increasingly moving to ensure that their products fulfill the criteria for “green” products [149,150]. A quick survey of the online marketplace Etsy (www.etsy.com) revealed a variety of containers and tableware with eco-friendly features. For example, a sandwich bag, typically single-use and made from plastic, is now also made from washable cotton fabrics. Another example is that of cup sleeves, formerly made only from paper, but now available in silicone, wool, wood, etc.

In recognizing this increase in consumer demand for environmentally-friendly items in the food and beverage industries, more research should be conducted to determine whether certain haptic properties of the container, tableware, and cutlery items can evoke differing consumer haptic perceptions. If such differences are found, additional research should be conducted to determine whether trends are consistent across all product types, i.e., solid foods, semi-solid foods, and beverages. While extensive research on the effects of different materials on consumer haptic perception and comfort has occurred in textile and apparel industries, very few studies in the food and beverage industries have been conducted on how materials affecting haptic properties of containers and tableware can influence consumer perceptions.

#### 3.3.2. Packaging

Packaging design has become an undeniably critical aspect of brand marketing [144,151]. In particular, the role of touch cues featured in product packaging, i.e., shape, texture, weight, and materials, is now deemed to be an important packaging component that could affect the consumer perception of the product contained [20,144,151]. As has been noted [20,151], touch cues of packaging components evaluated manually by consumers have been understudied compared to visual cues (e.g., colors and labels) because consumers typically use visual cues to develop expectations toward a product before touching its package [10,151]. 

Similarly to the situation of containers, tableware, and cutlery items, there have been very few studies on how packaging design could evoke different haptic perceptions. This may be due to limited technology access (e.g., 3D printing) in academia for creating packages with different haptic characteristics. Whatever the reason, there have been few previous studies describing how a package’s haptic characteristics could influence the haptic perception of consumers. However, with increasing consumer demand for eco-friendly packaging and more creative and novel packaging designs, this seems likely to become a topic of great interest [20,148,151].

## 4. Effects of Hand-Feel Touch Cues on Perceptions of Other Sensory Modules

Touching an object provides general information about its geometric (e.g., shape, size, orientation, and curvature) and material (e.g., temperature, compliance, texture, and weight) properties [152]. Touch sensations, especially textural sensations, derived from various sensory modalities can interact with one another, leading to an object’s overall touch perception [81]. Although cross-modal interactions of hand-feel touch cues with other sensory modality cues often occur over the span of purchasing or consuming food or beverage products, the study of such interactions has been under-evaluated. A summary of findings from a limited number of published articles related to cross-modal associations between hand-feel touch cues and other sensory modality cues is given in Table 1, Table 2, Table 3, Table 4 and Table 5. 

### 4.1. Visual Perception

For certain textural attributes related to shape judgment and dimension estimation, visual cues dominate touch cues, i.e., people tend to rely on information relayed from visual cues more than those from touch cues [12], but this is not always the case, and for textural attributes such as roughness, individuals rely more on touch cues than visual cues [153]. When an individual touches an object, the resulting sensation activates several regions in the brain that also respond to visual cues [154]. Among such regions, the lateral occipital complex (LOC) is considered to be one of the most-implicated because it is object-selective in both touch and vision [155]. The LOC has been shown to activate in response to both haptic [155] and tactile [156] stimuli. In addition to the LOC, since multiple loci along the intraparietal sulcus (IPS) are responsive to activities involving both visual and haptic discrimination of object features [157]. It is unsurprising that vision and touch senses can both be used to assess textural attributes such as roughness in abrasive papers [158]. Fenko et al. [12] reported that vision and touch were the most involved in both positive and negative product experiences, as well as being the most important senses used during food consumption [159]. However, the degree of sensory dominance between vision and touch depends greatly on the type of task [153] and, to date, most studies examining the effects of touch cues on visual perception have focused largely on cross-modal correspondences or synaesthesia.

Among the numerous studies on cross-modal associations, some have examined the association of touch perception with product attributes related to visual perception, such as color, luminance, and saturation. Ward et al. [160] demonstrated that low color luminance is closely associated with roughness and high pressure to the skin. In another study, Slobodenyuk et al. [161] associated high color luminance with high smoothness, high softness, high elasticity, and low adhesion. Conducting research in a more applicable setting, Tu et al. [142] evaluated consumer product expectations by examining food-product packaging using various materials, and found that organic glass was perceived as “bright”. As suggested by such studies, hand-feel touch perception can affect visual perception, which is also used in the judgment of product quality. The effects of cross-modal associations between hand-feel touch and visual cues must be considered very important in marketing, advertising, and product package design.

### 4.2. Auditory Perception

Neuroscience studies have shown that several regions in the brain are implicated in the multisensory integration of audio-tactile inputs [165]. In particular, the posterior superior temporal gyrus (pSTG), the adjacent posterior superior temporal sulcus (pSTS), and the left fusiform gyrus (FG) have been observed to become activated in response to multisensory object recognition across audition and touch [165,166,167]. However, the exact contribution of each sensory modality to the activation of these regions based on object recognition still remains unclear.

Earlier studies on cross-modal correspondence regarding touch and auditory cues have largely focused on the extent to which the sense of a word can be represented by its sound (“sound symbolism”), e.g., “bang” and “fizz” [145]. A study on sound symbolism revealed that participants judged high-pitched words like “mil” more than lower-pitched words like “mol” to better associated with a white or small object than with a black or large object [168,169]. This “sound-symbolism” notion can be translated into the cross-modal tendency for individuals to relate the haptic properties of an object to certain auditory properties. As demonstrated by several existing cross-modal correspondence studies that have successfully shown humans’ ability to associate tactile with audio attributes, people are generally inclined to associate lower pitch and quieter sounds with smoother, softer, and smaller objects, while higher pitch and louder sounds are more associated with rougher and larger objects [170,171,172,173,174]. 

With respect to food and beverage products, there has been growing interest in auditory product packaging design as more companies have come to recognize the power of sensory marketing. It has been observed that specific packaging-generated sounds can be associated with touch cues such as temperature [175,176]. Considering the rapid growth of interest in packaging design, this area should be further studied and companies should increasingly attempt to better incorporate cross-modality between touch and auditory cues in packaging design.

### 4.3. Olfactory Perception

It is widely known that flavor is a multisensory sensation comprised of sensations of taste, retronasal odor, and the oral somatosensory system [177]. Although previous studies have highlighted the influences of hand-feel touch cues on olfactory perception [178,179,180,181,182,183,184,185,186], this area of research remains understudied compared to the research area focusing on the effects of oral somatosensory cues on olfactory perception. Interestingly, research on the effects of touch cues on olfactory perception was spearheaded with studies related to wine tasting, possibly due to the common belief by wine connoisseurs that the shape of a wine glass could directly impact wine taste [180]. One of the more-studied aspects of wine consumption experience is the cross-modal effects of wine-glass shape (as evaluated manually) on the contained wines [187]. Glass shapes and dimensions were found to influence the aroma perception of the wines served, whether or not the participants were blindfolded [178,179,180,182]. While it has been proposed that such an effect of glass shape on odor perception could be due to the differences in the amount of wine exposed to environmental air [180], Russell et al. [188] revealed that participants could detect no difference between aerated wine and fresh wine samples served in the whole variety of glass shapes, although wine glass shape affected the composition of chemical compounds responsible for bitterness and astringency perceptions resulting from wine exposure to environmental air. It thus remains possible that there is an explanation yet to be discovered that could explain why the aroma perception of wine samples varies with respect to glass shape.

Several other studies have also investigated the effects of hand-feel touch cues on olfactory perception, although they highlighted only the effects on other types of beverages, not solid foods [185,186]. One such study found that cola drinks served in cola glasses were rated as more intense and pleasant than when served in other containers, i.e., a water glass or a bottle [185]. This was consistent with other studies that had investigated the congruency effects of the interaction between container (or packaging) and content [141,189]. In general, prior to the consumption of a product, through interaction with the container or packaging of a product, an individual may expert a certain experience, and when their expectation matches their consumption experience, it would be more likely that they perceive a greater liking of the product [190].

It is important to note that a majority of existing studies have not excluded visual effects during sample evaluation [180,185,186]. Since these studies have not isolated the sole effect of hand-feel touch cues on olfactory perception, further study is needed in this regard. Considering the established cross-modal relationship between olfactory and oral somatosensory sensations [191,192,193], it would be interesting to further explore the influences of hand-feel touch cues on olfactory perception in food and beverage settings, especially with respect to solid food samples.

### 4.4. Gustatory Perception

Unlike cross-modal studies on the effects of touch cues on olfactory perception, studies on gustatory perception have involved a wider variety of food and beverage products, including beverages such as beer [186,189,195], coffee [198], hot chocolate [189,199], cola drinks [185], and orange juice [189]; semi-solid foods such as yogurt [9,137,199], cream [200], and ice-cream [196]; and solid foods such as chips [197]. To elaborate on these cross-modal influences of hand-feel touch cues on gustatory sensations, the general consensus is that people associate certain features of packaging, tableware, and cutlery items with certain taste perceptions. In particular, angular, rough, or uneven items tend to be associated with foods and beverages of higher flavor intensity, bitterness, and saltiness, while round, smooth, or flat items tend to be associated with foods and beverages of lower flavor intensity and sweetness [9,186,196,197,198]. Other observations from existing studies show that when beverages are served in the containers they are generally expected to be served, i.e., when consumer expectations of consumption experience are matched with actual consumption experience, people tend to rate the beverages as being more pleasant and sweeter [137,185,195]. Hand-feel touch cues have also been found to influence food or beverage quality. For example, it was found that when participants were not allowed to touch the flimsy cup material, water was rated higher in quality [130].

Although the existing literature has revealed influences of touch cues on gustatory perception, such a cross-modal influence does not always occur. Slocombe et al. [177] found no cross-modal associations when the touch stimuli were presented in the form of the plateware (rough versus smooth plates) on which the food was served. Absence of cross-modal relationship between hand-feel touch and gustatory cues was also observed by Zhou et al. [194], who served noodles in bowls made of varying materials. This may indicate that the cross-modal association is stronger when both cues are presented together, i.e., not as separate stimuli, and it may also indicate a strong product-type effect [177,194]. It should also be noted that, with respect to studies on the effects of touch cues on olfactory perception, the results were potentially confounded by visual biases because participants were allowed to view the touch cues, representing one of the major challenges in conducting studies on the effects of hand-feel touch cues on taste perception.

### 4.5. Oral Somatosensory Perception

While the sense of touch can be perceived by various parts of the human body, the mouth and hands are generally the body parts used to sense and explore textural characteristics of products, especially food and beverage products. Note that tactile sensitivity does not necessarily indicate texture discrimination capability, an important aspect of food product evaluation [201]. Although there have been studies for determining whether differences exist between intra-oral and hand-feel touch sensitivities, the results have been mostly contentious. One example found the tongue to be slightly more sensitive in discriminating food texture, but no correlation between intra-oral and hand-feel sensitivities could be confirmed. In other words, a high level of intra-oral sensitivity does not necessarily signify a high level of hand-feel sensitivity [201]. Howes et al. [202] presented a variety of oral somatosensory cues using stimuli from “lolly sticks” made from different materials: polystyrene, rough polystyrene, stainless steel, copper, rough copper, birch, balsa, glass, or silicone. In that study, roughness was not considered to be a dominant textural sensation in oral texture evaluation, in contrast with studies on hand-feel evaluation where roughness was found to be the most dominant sensation [203]. While generally-dominant textural attributes in hand-feel touch evaluation are roughness, hardness, coldness, and slipperiness [203], a study by Howes et al. [202] found roughness to be less dominant than hardness and coldness. These suggest that certain body parts used for textural perception may be better at sensing particular textural attributes than others, e.g., roughness is better explored by hand-feel while hardness can be perceived equally well both orally and by hand-feel.

Hand-feel touch stimuli have been found to affect the oral somatosensory perception of food and beverage products [137,138,140,141,142,185,204,205,206,207,208]. The study conducted by Barnett-Cowan [204] showed, using pretzel samples, that perceived oral texture of a product can be modulated by the hand-feel touch perception of the same product. In this study, half of the participants were presented with half-stale, half-fresh pretzels, while the other participants were presented with either whole fresh or whole stale pretzels. Blindfolded participants were then asked to hold one half of the pretzel while orally evaluating the other half. Fresh pretzel tips were perceived to be staler and softer when participants were holding the stale pretzel end, and vice versa. The same “mirror” effect was also observed in non-edible products [209]. The cross-modal influence of hand-feel touch cues on oral somatosensory perception can also be observed for hand-feel touch stimuli from packaging, tableware, and cutlery items. Biggs et al. [206] found that biscuits were rated crunchier and rougher when served on rougher-surfaced plates than on smoother-surfaced plates. This trend of sensation transference for rougher-surfaced versus smoother-surfaced containers was not only observed for solid (e.g., biscuits) foods, but also for semi-solid (e.g., yogurt) foods [205]. In another study, Piqueras-Fiszman and Spence [205] found that biscuits were rated as crunchier and harder when they were presented in a container with a rough sandpaper finish than when presented in a smooth-coated container. However, in their study the ratings of oral textural attributes of yogurt samples were not influenced by the textural attributes of yogurt-sample containers, although tableware weight had an impact on the oral textural attributes; when a yogurt sample was presented in a heavier bowl, participants rated the yogurt as denser than when presented in lighter bowls [139,140].

Cutlery items have also been found to influence certain textural attributes of food or beverage samples. In contrast to the results of a previous study where yogurt presented in heavier bowls was rated denser (as well as more expensive) than in lighter bowls [139,140], yogurt consumed using lighter spoons were rated as denser than that consumed using heavier spoons [137]. Harrar and Spence [137] proposed that this discrepancy with earlier studies in cross-modal correspondence trends [139,140] was due to the participants’ expectation with respect to tableware weight. In other words, when consumer tableware-weight expectations are confirmed by actual tableware experience, the tasted food sample would be perceived as better, i.e., denser and more expensive. 

Variation in packaging or container materials could result in differences in oral somatosensory perceptions. McDaniel and Baker [210] showed that potato-chip crunchiness was rated higher when they were packed in polyvinyl bags rather than wax-coated paper bags, illustrating the effects of packaging materials on the textural perception of content. The follow-up blind study revealed no significant bag-dependent differences in potato chips, further confirming the idea that packaging properties can alter the oral textural perception of food [210]. In that study, the packaging material may have been associated with certain semantic and/or affective meanings or connotations that, in turn, could have influenced consumer perception of the packaging content. This tendency of individuals to relate and combine connotations from multiple sensory modalities, i.e., textural cues from packaging and textural product qualities, was further demonstrated in the area of product packaging by a word-association study conducted by Ares and Deliza [211]. Although their study involved no direct physical touching of the packaging, participants semantically associated round packaging shapes with product textural attributes such as “runny”, “creamy”, and “soft” milk desserts, while square (more angular) packages were associated more with “thick” and “low-calorie” milk desserts, resulting in higher desirability of milk desserts served in round packages [211]. It is important to note that these studies show that product ratings generally follow the ratings of the packaging, tableware, and cutlery items, similar to Schifferstein’s results [141].

Hand-feel touch cues also influence the pleasantness of oral somatosensory sensations. Still and carbonated water samples were rated as more pleasant and less carbonated when served in plastic cups (versus sandpaper and satin-covered cups) that were lighter (versus heavier) [138,208]. From these studies, it can be seen that hand-feel touch cues influence mouthfeel or oral trigeminal sensations, i.e., carbonation burns (see also [185,207]). Weight, another component of haptic sensations, has also been shown to bias consumer perception of oral somatosensory perception. As described earlier, the weights of tableware and cutlery items do not seem to reflect the same influence on oral somatosensory perceptions [137,140].

## 5. Effects of Hand-Feel Touch Cues on Consumer Emotion and Behavior

### 5.1. Consumer Emotions

Several different explanations related to how or why product characteristics can evoke emotions have been proposed. One suggested theory proposed by Desmet [17] is referred to as an “appraisal approach”. Emotions and consequent emotion-regulated behaviors can play a role in indicating the well-being of individuals with respect to their relationship with their surroundings [17]. For individuals to feel emotions, they must grasp the situational meaning of perceived changes occurring in their interactions with their surroundings and how these changes could influence their well-being, i.e., the individuals must appraise such an occurrence’s importance to their welfare. This appraisal differs among individuals since it acts as an intermediate stage between an event and resulting emotions, and different individuals experiencing the same event may perceive it differently and experience different emotions as a result [17]. According to Ortony et al. [18], there are three different types of appraisal: usefulness, pleasantness, and rightfulness, and they combine to assist individuals in determining whether a perceived change in surroundings is beneficial to their well-being. For example, one situation can elicit positive emotions because it is perceived to be useful, pleasurable, or rightful, while, in contrast, negative emotions can be elicited from a situation perceived to be harmful, painful, or wrongful. These appraisals are also closely connected to the individual’s prior experience to the product. It has been shown that the hand-feel properties of a product can be remembered 1 week after only a short exposure of 10 s to it [212]. This concurs with extensive research on mere exposure effect in the domain of vision, where mere exposure to a stimulus enhances the preference towards it [213,214]. This mere exposure effect in the domain of hand-feel touch has also been found to potentially follow a common cognitive basis [215,216]. Empirical research on the effect of touch on interpersonal behavior has shown that, depending on the context, touch communicates either positive emotional intentions (e.g., warmth and intimacy) or negative emotional intentions (e.g., pain or discomfort), and touch can also augment emotional effects from other sensory modalities [15,16].

The appraisal approach used by an individual to form such emotional behavior also applies to his or her valuation and appraisal of a product. The emotional influence of a product on an individual is dependent on “*its material qualities, purposes, meanings, expressions, and on what it does or fails to do*” [17]. To assess the success of a product design, an individual must physically touch the product, and the physical features and tangible qualities of a product, such as weight, texture, and surface, can considerably influence a consumer’s appreciation of its value; it may be observed as a source of affective pleasure and contribute to a wholesome experience of human-product interaction [18].

In addition to evoking emotional associations with the textural attributes of a product contained within, product packaging design could also be associated with specific affective connotations. Chen et al. [217] showed that thermally-warm materials were considered to be “natural”, but not "exciting" or "precious". In general, people have a tendency to prefer smooth surfaces over rough ones [81], but it should be noted that consumers tend to consider not only affective experiences, but also functionality and other abstract connotations. During the lexicon development phase for the evaluation of bottled blackcurrant beverages by Ng et al. [2], they found that consumers generally describe packaged products using descriptors from emotional, abstract, and functional classifications, highlighting that consumers can also place particular emphasis on the packaging functionalities whenever they evaluate packaging at first glance. Therefore, when designing packaging, companies need to consider how it can be functionally beneficial, e.g., maintain product freshness, prolong shelf-life, be easy to open, etc., while also incorporating haptic features that could enhance consumer perception of product quality and attributes.

### 5.2. Consumer Purchase Behavior

Because touch cues acquaint consumers with the material properties of a product, including information about texture, weight, and temperature, consumers can also focus on product quality and value [218,219]. This is why, especially at the point of sale, consumers may be more motivated to touch a product to assess its quality. The more variety in one or more of these material attributes, i.e., texture, weight, and/or temperature, the more likely that a consumer will be motivated to touch it for purposes of product judgment [220]. In one study [220], products with the greatest variation in material properties were touched longer than those with lesser variation and lesser means of evaluation. It has also been established that when consumers are allowed to physically touch products for examination, products using varied materials are most likely to be preferred [11]. However, if the shopping environment does not allow for haptic exploration, as in the case of online shopping, verbal description of the textural properties can effectively compensate for a lack of touch [221]. Moreover, consumer perceptions of ownership and valuation of an object can be modulated with mere touching or imagery encouraging touch in which participants, after actively interacting with an object, were asked to imagine whether they could take it home [222]. Consumers develop expectations of food products with respect to sensory attributes at the point of product appraisal, involving visual and/or touch evaluation of product packaging [223]. If the expectations are not subsequently validated by the sensory qualities of the product, consumer disconfirmation may occur, resulting in a change of product quality perception and purchase behavior [224]. Confirmation and disconfirmation can be associated with four consumer-behavior possibilities: (1) assimilation (ratings move toward expectations), (2) contrast (ratings move away from expectations), (3) generalized negativity (ratings decrease under all conditions of disconfirmation); and (4) assimilation-contrast (at low disconfirmation, an assimilation effect occurs, while at high disconfirmation, a contrast effect occurs) [2,224]. Confirmation of consumer expectations through sensory attribute evaluation usually results in repeated product purchase, highlighting the importance of studies regarding the effects of both intrinsic sensory attributes and extrinsic touch cues of product packaging.

## 6. Applications to Food and Beverage Industries and Future Research

As an increased acknowledgment of the effects of touch cues on consumer perception, liking, and behavior of food products has occurred, there has been an increase in the number of business and research efforts aimed at designing and producing creative packaging designs that incorporate haptic components. Spence and Gallace [144] emphasized that recent technological developments have generated novel packaging designs at a cheaper cost and a faster rate. Hand-feel touch cues provide information about the material properties of a product (e.g., its texture, softness, weight, and temperature, etc.) [218,219]. McCabe and Nowlis [11] showed that products with more varied materials are more likely to be preferred by consumers. Because of the dominance of tactile over visual cues with respect to product evaluation and liking in some contexts [225], designing appealing product packaging that motivates consumers to touch it would be greatly advantageous in the competitive food and beverage market. 

The rapid development of technologies in the current era (e.g., 3D printing) makes it even more possible to create novel and interactive packaging designs that would assimilate more consumer engagement in the hope of increasing product purchase. In fact, Van Rompay et al. [198] demonstrated that the application of 3D printing technology to cup design could influence the taste perception of the beverages they contain. With the continuous exploration of the potential influences of hand-feel touch cues as part of the packaging, tableware, and cutlery designs on consumer consumption experience and product perception, more creative and novel tableware and cutlery items should be expected in the near future.

Although food and beverage professionals are encouraged to integrate oral and hand-feel touch cues into the product consumption experience, it should be noted that the effect of product type with cross-modal correspondence involving touch could affect the consumer perception of other sensory modalities and purchase behavior [226]. A majority of cross-modal correspondence studies, especially those regarding oral and hand-feel touch cues, noted that the findings of specific studies usually cannot be generalized to other product types [138,226]. Researchers should, therefore, continue studies on various solid, semi-solid, and liquid foods to develop sufficient evidence of cross-modal associations with touch before generalizing conclusions. Furthermore, a majority of cross-modal correspondence research has neglected the possible effects of the intensity of each sensory modality evaluated. In general, cross-modal research focuses on association (i.e., best-match question), not addressing the intensity of each cue, e.g., impacts of high intensity of cue A on low intensity of cue B versus impacts of high intensity of cue A on high intensity of cue B. This may either enhance or suppress the extent to which cross-modal correspondence can influence an individual. Additionally, while certain personal tendencies, such as those measured by the NFT scale and certain neuropsychological factors, have been found to modulate the degree to which an individual is affected by cross-modal correspondences involving touch, there may be other factors that also regulate these effects. Notably, there have been very few published studies on the cross-modal correspondence between trigeminal hand-feel touch and trigeminal oral sensations. Previous studies that have highlighted the cross-modal association between touch and trigeminal cues have mainly focused on carbonation feelings [138,185,207,227]. Individuals who are 6-*n*-propylthiouracil (PROP) supertasters tend to perceive certain oral irritant stimuli, e.g., capsaicin, piperine, and ethanol, at greater intensities than non-tasters [228], suggesting that genetic factors may also play a role in modulating the effect of cross-modal association. 

Further research is needed to determine whether cross-modal correspondences between hand-feel touch and other sensory modalities cues are implicit or based on learned experiences. The use of neuroscience techniques, such as electroencephalogram (EEG) or other procedures that measure brain activities, may need to be considered for assessment. Ultimately, there is an abundance of opportunities for further research in the field of touch cues. There are many modulating factors that remain unknown, as well as reasons and mechanisms to explain why cross-modal correspondence between touch and other sensory modalities, emotions, and consumer behavior occur. Despite the many unexplored topics in the field, it is obvious that incorporating more hand-feel and oral touch cues related to both intrinsic and extrinsic aspects of food and beverages could elevate a product above its competitors, especially in an increasingly and rapidly dynamic and competitive market.

## 7. Conclusions

The effects of hand-feel touch cues, although largely underestimated in the past, are now increasingly acknowledged by food and beverage professionals. This review provides substantial evidence accounting for such a trend in the food industry, although the identification of exact mechanisms underlying the effect of hand-feel touch cues on consumer perception and experience of food and beverages remain elusive. More specifically, the incorporation of appropriate haptic components into the consumption experience of food and beverage products can induce positive influences on consumer perception, liking, emotions, and purchase behavior. Notably, such hand-feel touch cues can be presented in a variety of ways that include food-product surfaces, tableware and cutlery items, containers, packaging, and surrounding contexts (e.g., on the dining table or on supermarket shelves). There are also plentiful opportunities for further research in the field of cross-modal associations of hand-feel touch cues with other sensory modalities, and these are especially motivated by the relatively few studies in this field, compared to those related to the effects of oral somatosensory cues on other senses. Moreover, currently-available 3D printing technology, haptic technology, and immersive technology can help product developers, designers, sensory professionals, and marketers creatively incorporate various haptic components into their products, thereby enriching consumer experience and satisfaction and increasing product-market competitiveness.

## Figures and Tables

**Figure 1 foods-08-00259-f001:**
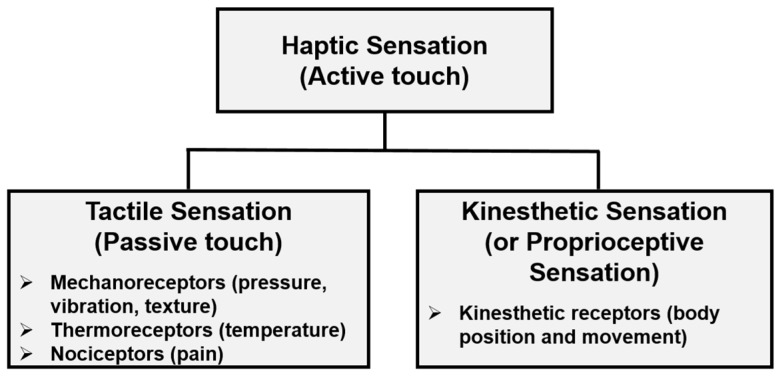
The concepts of terminologies commonly used in the literature associated with a sense of touch.

**Table 1 foods-08-00259-t001:** The summary of findings regarding cross-modal associations between visual and hand-feel touch cues.

Types of Visual Cues	Presentation Types of Visual Cues	Types of Touch Cues	Presentation Types of Touch Cues	Key Findings	References
Hue (black/white)	Colored squares (via computer)	Vibrotactile	Computer-controlled shaker	Low-frequency vibrations were associated with a black hue; high-frequency vibrations were associated with a white hue	Martino & Marks [162]
Hue (red/white wine)	Wine color	Weight	Wine bottles	Red wine bottles were rated heavier compared to white wine	Piqueras-Fiszman & Spence [163]
Luminance, chroma, hue	Color wheel (via computer)	Temperature, roughness, vibrotactile, pressure	Sandpaper (roughness), solenoid tapper (vibrotactile)	Low color luminance was associated with roughness and high pressure to skin	Ward et al. [160]
Luminance, chroma, hue	Color wheel (via computer)	Hardness/softness, pointed/roundness, roughness/smoothness	Foam cubes (hard-soft), wooden 3-D shapes (pointed-round), sandpaper-covered flat surfaces (rough-smooth)	High luminance correlated with high softness and roundness; high chroma correlated with smoothness and softness; specific color hues were associated with certain tactile sensations	Ludwig & Simner [164]
Luminance, chroma, hue	Color wheel (via computer)	Hardness, roughness, heaviness, elasticity, adhesiveness	Programmed haptic device (SensAble PHANTOM OMNI^®^)	High color luminance was associated with high smoothness, high softness, high elasticity, and low adhesion	Slobodenyuk et al. [161]

**Table 2 foods-08-00259-t002:** The summary of findings regarding cross-modal associations between auditory and hand-feel touch cues.

Types of Auditory Cues	Presentation Types of Auditory Cues	Types of Touch Cues	Presentation Types of Touch Cues	Key Findings	References
Loudness, pitch	(Modified) sounds of participants rubbing their own palms together played back to the participants	Roughness/moistness, dryness/smoothness	Participants’ own skin (participants rubbing their palms together)	Increased sound intensity and high pitch were more associated with higher smoothness/dryness of human palmar skin	Jousmäki & Hari [174]
Loudness, pitch	(Modified) sounds of participants touching the touch stimuli played back to the participants	Roughness	Abrasive closed-coat silicon carbide papers attached on plastic discs	Decreased sound intensity and lower pitch increased the perception of tactile smoothness	Guest et al. [170]
Loudness, auditory associations	Recorded sounds	Roughness	Programmed haptic device (SensAble PHANTOM)	Rougher textures were correlated with increased sound intensity; smoother textures were more associated with decreased sound intensity	Peeva et al. [171]
Loudness, pitch, sound type (violin vs. flute), auditory associations	Recorded sounds	Sharpness/bluntness, roughness/smoothness, hardness/softness, weight, temperature	Touch-related terms (i.e., no physical touch stimuli)	High smoothness and softness can be associated with low sound intensity, low pitch, and flute sound (compared to violin), while high sharpness can be associated with high sound intensity and flute sound (compared to violin)	Eitan & Rothschild [172]
Pitch, auditory associations	Daniel Barenhoim’s recording of Beethoven’s piano sonata (2^nd^ movement, opus 111)	Temperature, hardness/softness, weight, roughness/softness, sharpness/bluntness, size (small/large), thinness/thickness	Touch-related terms (i.e., no physical touch stimuli)	High pitch was more associated with “small”, “thin”, “sharp”, “smooth”; low pitch was more associated with “large”, “thick”, “heavy”, “blunt”, “rough”	Eitan & Timmers (Experiment 2) [173]

**Table 3 foods-08-00259-t003:** The summary of findings regarding cross-modal associations between olfactory and hand-feel touch cues.

Types of Olfactory Cues	Presentation Types of Olfactory Cues	Types of Touch Cues	Presentation Types of Touch Cues	Key Findings	References
Orthonasal odor	Wine (red & white); Overall aroma intensity, fruity aroma intensity	Shape	Wine glasses	Aroma intensities were rated higher when wines were served in bowl-shaped glass than in tulip-shaped glass (in white and red wines)	Cliff [178]
Retronasal odor	Hot chocolate, beer, & orange juice; Overall flavor intensity, overall pleasantness	Shape	Receptacle (bottles vs. cups vs. glasses)	Hot chocolate, beer, and orange juice were rated to be most pleasant when consumed from bottles (compared to glasses and cups)	Raudenbush et al. [189]
Orthonasal odor	Wine (red); Overall aroma intensity, fruity aroma intensity, vinegar aroma intensity, oak/woodiness aroma intensity, mustiness aroma intensity	Shape	Wine glasses	Odor intensity of red wine samples were rated as less intense when presented in tapered bulb-shaped glasses than open bulb-shaped and square-shaped glasses	Delwiche & Pelchat [179]
Retronasal odor	Wine (red & white); Overall aroma intensity, overall pleasantness	Shape	Wine glasses	Odor intensity of red and white wine samples were rated as most intense when presented in bulbous-shaped glass than tulip-shaped and beaker-shaped glasses	Hummel et al. [180]
Orthonasal odor	Lemon & animal odors	Roughness/softness	Treated fabric squares	Fabrics of varying degrees of softness were rated softer in the presence of a lemon odor (compared to an animal-like odor)	Demattè et al. (Experiment 1) [181]
Orthonasal odor, retronasal odor	Wine (toasted odor wine); Overall aroma intensity, overall quality	Shape	Wine glasses	Odor intensity of toasted wine samples were rated as most intense when presented in a specific wine glass (Schott Zwiesel type Cask-aged spirits 8432/17 with 209 x 76 mm dimensions)	Vilanova et al. [182]
Orthonasal odor	Feminine fragrance (Hanae Mori White) & masculine fragrance (Hanae Mori Black) (Experiment 1); Pumpkin cinnamon & eucalyptus-spearmint (Experiment 2); Pleasantness, likeability	Roughness/smoothness (Experiment 1); Temperature (Experiment 2)	Textured paper (Experiment 1); Gel packs (warm & cold) (Experiment 2)	Experiment 1: Smooth-textured paper was rated more positively in the presence of a feminine smell; rough-textured paper was rated more positively in the presence of a masculine smellExperiment 2: A warm gel-pack with a “warm” pumpkin cinnamon smell was rated more positively than with a “cold” eucalyptus-spearmint smell; a cold gel-pack with a “cold” eucalyptus-spearmint smell was rated more positively than a “warm” pumpkin cinnamon smell	Krishna et al. (Experiments 1 & 2) [183]
Retronasal odor	Lemon yogurt; Overall flavor intensity	Curvature (round/angular)	Yogurt packaging/container	Angular yogurt containers were perceived as more intense in taste (compared to rounded yogurt containers)	Becker et al. [9]
Orthonasal odor	Liquid soap; Overall fragrance intensity	Weight	Soap bottles	Fragranced liquid soap in heavier bottles were rated as having a higher fragrance intensity than soap in lighter bottles	Gatti et al. [184]
Retronasal odor	Noodles; Savory flavor intensity	Shape, material	Plates, bowls (ceramic, glass, paper, metal)	No differences with regards to touch stimuli	Zhou et al. (Experiment 2) [194]
Retronasal odor	Beer; Overall flavor quality, pleasantness	Shape, material	Beer cans vs. bottles	Beers served in bottles were rated higher in taste quality (poor/good) (compared to cans)	Barnett et al. [195]
Orthonasal odor, retronasal odor	Cola & sparkling water; Overall aroma intensity, pleasantness	Shape	Glasses	The aromas of cola drinks served in cola glass were rated more intense and pleasant than when served in a straight water glass or bulbous bottle	Cavazzana et al. [185]
Orthonasal odor, retronasal odor	Beer; Overall aroma pleasantness, overall flavor pleasantness, overall flavor intensity; fruitiness aroma intensity	Shape	Glasses	Higher glass curvature was associated with higher overall odor intensity (in beer)	Mirabito et al. [186]
Retronasal odor	Ice cream; Overall flavor intensity	Sharpness/smoothness	3D-printed cups	Ice cream served in angular-surfaced bowls were rated higher in intensity	Van Rompay et al. [196]
Retronasal odor	Potato chips; Overall flavor intensity	Roughness/smoothness	Bowls	Salted chips served in rough and uneven bowls were rated higher in saltiness and taste intensity than when served in smooth and even bowls	Van Rompay & Groothedde [197]

**Table 4 foods-08-00259-t004:** The summary of findings regarding cross-modal associations between gustatory and hand-feel touch cues.

Types of Gustatory Cues	Presentation Types of Gustatory Cues	Types of Touch Cues	Presentation Types of Touch Cues	Key Findings	References
Sweetness, bitterness, sourness, saltiness	Wine (red & white); Taste intensity	Shape	Wine glasses	Red and white wine samples were rated as more sour in beaker-shaped glasses	Hummel et al. [180]
Bitterness	Lemon yogurt; Taste intensity	Curvature(round/angular)	Yogurt packaging/container	No differences	Becker et al. [9]
Sweetness, bitterness, sourness, saltiness	Cream; Taste intensity	Cutlery item material	Spoons	Spoons of different materials could transfer certain tastes and enhance the dominant taste of cream samples; Copper and zinc spoons lent a degree of bitterness and metallic flavor to the cream	Piqueras-Fiszman et al. [200]
Sweetness (Experiment 1); Saltiness (Experiment 3)	Yogurt (Experiment 1); Cheese (Experiment 3); Taste intensity, pleasantness	Cutlery item weight and size (Experiment 1); Cutlery item type (Experiment 3)	Spoons (Experiment 1); Cutlery items (toothpicks vs. cheese knives vs. spoons)	Experiment 1: Yogurt was rated as sweeter when served with the smallest spoons (compared to larger spoons)Experiment 3: Cheese was rated as saltier when sampled using a knife (compared to spoon, toothpick, and fork)	Harrar & Spence (Experiments 1 & 3) [137]
Sweetness, bitterness, sourness	Cold tea	Material	Cups (glass, plastic, paper)	No differences with regards to touch stimuli	Tu et al. (Experiment 1) [142]
Sweetness	Noodles	Shape, material	Plates, bowls(ceramic, glass, paper, metal)	No differences with regards to touch stimuli	Zhou et al. (Experiment 2) [194]
Sweetness, bitterness, sourness, saltiness	Cola & sparkling water; Taste intensity, pleasantness	Shape	Glasses	Cola drinks served in a cola glass were perceived to be sweeter and more pleasant than when served in a water glass or bulbous bottle	Cavazzana et al. [185]
Sweetness, bitterness	Beer; Taste intensity	Shape	Glasses	Higher glass curvature was associated with a higher fruitiness (in beer)	Mirabito et al. [186]
Sweetness, bitterness	Hot chocolate & coffee; Taste intensity, overall liking	Curvature (round/angular)	3D-printed cups	Drinks served in angular-surfaced cups were rated higher in bitterness and intensity; Drinks served in rounder-surfaced cups were rated higher in sweetness and lower in intensity (in hot chocolate and coffee)	Van Rompay et al. [198]
Sweetness, sourness	Ice cream; Taste intensity	Sharpness/smoothness	3D-printed cups	Ice cream served in smoother-surfaced bowls were rated higher in sweetness; No differences on sourness	Van Rompay et al. [196]
Saltiness	Potato chips; Taste intensity	Roughness/smoothness	Bowls	Salted chips served in rough and uneven bowls were rated higher in saltiness and taste intensity than when served in smooth and even bowls	Van Rompay & Groothedde [197]

**Table 5 foods-08-00259-t005:** The summary of findings regarding cross-modal associations between oral and hand-feel touch cues.

Types of Oral Touch Cues	Presentation Types of Oral Touch Cues	Types of Touch Cues	Presentation Types of Touch Cues	Key Findings	References
Crispness	Potato chips; Attribute intensity	Material	Packaging bags (polyvinyl vs. wax-coated)	Potato chips in polyvinyl bags were perceived to be crisper	McDaniel & Baker [210]
Weight, thinness/thickness, softness/hardness, temperature, roughness/smoothness, flexible/stiff	Hot tea & carbonated beverage; Attribute intensity	Weight, thinness/thickness, softness/hardness, temperature, roughness/smoothness, flexible/stiff	Cups (of varying materials); Attribute intensity	Product ratings for certain attributes (e.g., warmness and softness), followed packaging ratings for those attributes	Schifferstein (Experiments 1 & 2) [141]
Softness/firmness, freshness/staleness	Pretzels; Attribute intensity	Softness/firmness, freshness/staleness	Pretzels; Attribute intensity	Stale pretzels evaluated by hands were associated with a staler and softer perception of fresh pretzels evaluated orally; Fresh pretzels evaluated by hands were associated with a fresher and firmer perception of stale pretzels evaluated orally	Barnett-Cowan [204]
Density	Yogurt; Attribute intensity	Weight	Bowls	Yogurt served in heavier bowls were rated as denser and liked more than when served in lighter bowls	Piqueras-Fiszman & Spence [140]
Crunchiness	Biscuits; Attribute intensity	Roughness/smoothness	Containers	Biscuits served in rough-finished containers were rated as crunchier than when served in smooth-coated containers	Piqueras-Fiszman & Spence [205]
Density	Yogurt; Attribute density	Cutlery item weight	Spoons	Yogurt sampled using lighter spoons was rated as denser and more expensive than when sampled using heavier spoons	Harrar & Spence (Experiment 1) [137]
Carbonation	Still & carbonated water; Attribute intensity, pleasantness	Weight	Cups (plastic)	Still and carbonated water samples were rated as less pleasant and more carbonated when served in heavy plastic cups (compared to lighter plastic cups)	Maggioni et al. [138]
Temperature	Tea; Attribute intensity	Material	Cups (glass, plastic, paper)	Tea samples served in glass cups were perceived to be colder (compared to plastic and paper cups)	Tu et al. [142]
Crunchiness, roughness	Biscuits; Attribute intensity	Roughness/smoothness	Plates	Biscuits served in rougher-surfaced plates were rated as crunchier and rougher than when served in smoother-surfaced plates	Biggs et al. [206]
Carbonation	Cola & water; Attribute intensity	Shape	Glasses	Cola and water served in a bulbous bottle were perceived to have more carbonation than when served in cola or water glasses	Cavazzana et al. [185]
Carbonation	Fruit drinks; Attribute intensity	Weight	Cups (plastic)	Highly bitter fruit drinks were perceived to be more carbonated when presented with heavier plastic cups (compared to lighter plastic cups)	Mielby et al. [207]
Freshness, lightness	Still & carbonated water; Attribute intensity, pleasantness	Roughness/smoothness	Cups (plain, sandpaper-covered, satin-covered)	Still and carbonated water samples were more pleasant, fresher, and more light when served in plastic cups (compared to sandpaper and/or satin-covered cups)	Risso et al. [208]
Crispness	Potato chips; Attribute intensity	Roughness/smoothness	Bowls	No differences	Van Rompay & Groothedde [197]

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
