# Peer review of "Hand-Feel Touch Cues and Their Influences on Consumer Perception and Behavior with Respect to Food Products: A Review"

_foods, 2019, doi:10.3390/foods8070259_

Round 1
Reviewer 1 Report
1) This paper is really interesting, however I think it is too long (referring to some parts). In my opinion you should reduce/rebuild/shorten the part referring to the perception of touch cues – particularly lines 121- 165 (pp 3-4).
2) Moreover, among factors influencing hand-feel touch perception you mentioned 3 groups of them (lines 210-211 pp 5), in my opinion it will be better to use the same name according the third group of factors (external interface related) in point 3.3. line 315, pp 7.
3) You can also change the size of fonts in the tables (it is worth to make them smaller).
4) Please check the paper carefully, because there are some minor mistakes in it (for example):
Line 368, page 8 – summary
Line 584, page 23 – depending on
Author Response
Reviewer 1
Q1) This paper is really interesting, however I think it is too long (referring to some parts). In my opinion you should reduce/rebuild/shorten the part referring to the perception of touch cues – particularly lines 121- 165 (pp 3-4).
Answer>>
Thank you for your suggestion. However, an overview about anatomical and physiological aspects of touch perception would be essential in this review, providing a better understanding of how individuals perceive touch stimuli. Therefore, the authors would like to maintain the paragraphs. Thank you for your kind understanding in advance.
Q2) Moreover, among factors influencing hand-feel touch perception you mentioned 3 groups of them (lines 210-211 pp 5), in my opinion it will be better to use the same name according the third group of factors (external interface related) in point 3.3. line 315, pp 7.
Answer>>
Thank you for your kind suggestion. As the Reviewer 1 suggested, the title of section 3.3. was changed to “External interface-related factors” (line 350).
Q3) You can also change the size of fonts in the tables (it is worth to make them smaller).
Answer>>
Thank you for your kind suggestion. The size of fonts in the tables has been determined by a journal publisher.
Q4) Please check the paper carefully, because there are some minor mistakes in it (for example):
Answer>>
The authors carefully checked and corrected typos and grammar errors in the manuscript.
Q5) Line 368, page 8 – summary
Answer>>
It was corrected to “summary” (line 405).
Q6) Line 584, page 23 – depending on
Answer>>
It was corrected to “depending on” (line 627).
Reviewer 2 Report
This paper is a review describing the effects of hand-feel touch cues on consumer perception and behavior on food products. It is a very comprehensive work. and it needs some corrections.
Section1 - Introduction: The introduction of this paper shows the background information regarding the interaction between the touch cues sensations and the hedonic and emotional responses of consumers. The introduction is well constructed in the sense that it shows some general information at the beginning and then it lays out the arguments and justifications of the importance of this review. However, it seems that the focus of this introduction is for products that have packages or are enclosed in a package. There are some retail products (fruits, vegetables) that do not necessarily have a package but the touch cues sensations are still very important for the sensorial assessing those products. It will be useful to discuss this point in the introduction.
Section 2 – Perception of touch cues: It is a comprehensive section regarding the types of mechanoreceptors and their functions in the touch perception. However, from the introduction section, it was established that kinaesthetic sensations can also play an important role in the haptic perception system. It may be useful to include a description of the perception mechanisms (muscle, joints, special perception) employed in the kinaesthetic sensations.
Section 3 – Factors: I think that section 3.1 (Product-related factors) is too short compared to the other subsections in this part. This is an important topic as the differences in the products’ properties can lead to different touching experiences. It can be mentioned that foods either touched by using packaging or directly touch can produce different consumers’ behaviors. It may be useful to include the link of the other products properties as well (aroma, visual) since those can interact with the touch experience in the overall perception of the product (although this topic is covered in section 4, a brief introduction of the other product properties and their interaction with the touch sensation can be included).
Section 5 – Emotion and Behaviour: It may be useful in this section to refer back to the three main dimensions that describe emotions (valence, arousal, and motivation). Similar to other perceptual sensations (hearing, visual, taste, smell), the particular stimuli can elicit a particular set of emotions (given also the previous associations of the stimuli with the subject). In the case of the touching experience, memory and familiarity play an important role in the emotional responses of consumers. This can be discussed in this section.
Line 584: It should be “depending”
Author Response
Reviewer 2
Q1) This paper is a review describing the effects of hand-feel touch cues on consumer perception and behavior on food products. It is a very comprehensive work and it needs some corrections.
Answer>>
Thank you so much for your generous comments.
Q2) Section1 - Introduction: The introduction of this paper shows the background information regarding the interaction between the touch cues sensations and the hedonic and emotional responses of consumers. The introduction is well constructed in the sense that it shows some general information at the beginning and then it lays out the arguments and justifications of the importance of this review. However, it seems that the focus of this introduction is for products that have packages or are enclosed in a package. There are some retail products (fruits, vegetables) that do not necessarily have a package but the touch cues sensations are still very important for the sensorial assessing those products. It will be useful to discuss this point in the introduction.
Answer>>
Thank you for your great and helpful comments. As the Reviewer 2 suggested, the importance of touch cues on consumption/evaluation of unpacked food products such as fruits and vegetables was addressed (lines 31-42).
Q3) Section 2 – Perception of touch cues: It is a comprehensive section regarding the types of mechanoreceptors and their functions in the touch perception. However, from the introduction section, it was established that kinaesthetic sensations can also play an important role in the haptic perception system. It may be useful to include a description of the perception mechanisms (muscle, joints, special perception) employed in the kinaesthetic sensations.
Answer>>
Thank you for wonderful comments. As the Reviewer 2 suggested, some background about the mechanisms underlying the kinaesthetic sensations was added (lines 155-164). Since the Reviewer 1 showed a concern about the lengthy manuscript, it was briefly addressed.
Q4) Section 3 – Factors: I think that section 3.1 (Product-related factors) is too short compared to the other subsections in this part. This is an important topic as the differences in the products’ properties can lead to different touching experiences. It can be mentioned that foods either touched by using packaging or directly touch can produce different consumers’ behaviors. It may be useful to include the link of the other products properties as well (aroma, visual) since those can interact with the touch experience in the overall perception of the product (although this topic is covered in section 4, a brief introduction of the other product properties and their interaction with the touch sensation can be included).
Answer>>
Thank you for helpful comments. As the Reviewer 2 suggested, the Section 3.1. was further revised (lines 233-258).
Q5) Section 5 – Emotion and Behaviour: It may be useful in this section to refer back to the three main dimensions that describe emotions (valence, arousal, and motivation). Similar to other perceptual sensations (hearing, visual, taste, smell), the particular stimuli can elicit a particular set of emotions (given also the previous associations of the stimuli with the subject). In the case of the touching experience, memory and familiarity play an important role in the emotional responses of consumers. This can be discussed in this section.
Answer>>
Thank you for great comments. As the Reviewer 2 suggested, some factors influencing emotions related to touch cues were addressed (lines 620-626).
Q6) Line 584: It should be “depending”
Answer>>
It was corrected to “depending” (line 627).